# Molecular Epidemiology and Evolution of Coxsackievirus A14

**DOI:** 10.3390/v15122323

**Published:** 2023-11-26

**Authors:** Liheng Yu, Qin Guo, Haiyan Wei, Yingying Liu, Wenbin Tong, Shuangli Zhu, Tianjiao Ji, Qian Yang, Dongyan Wang, Jinbo Xiao, Huanhuan Lu, Ying Liu, Jichen Li, Wenhui Wang, Yun He, Yong Zhang, Dongmei Yan

**Affiliations:** 1National Polio Laboratory, WHO WPRO Regional Polio Reference Laboratory, National Health Commission Key Laboratory for Biosecurity, National Health Commission Key Laboratory of Medical Virology, National Key Laboratory of Intelligent Tracking and Forecasting for Infectious Diseases, National Institute for Viral Disease Control and Prevention, Chinese Center for Disease Control and Prevention, Beijing 102206, China; yuliheng2021@163.com (L.Y.); 2470453013@163.com (Q.G.); zhusli@126.com (S.Z.); jitj@ivdc.chinacdc.cn (T.J.); yangqian@ivdc.chinacdc.cn (Q.Y.); wangdy@ivdc.chinacdc.cn (D.W.); mr_mint1114@sina.com (J.X.); luhuanhuan0908@163.com (H.L.); ilene1996@163.com (Y.L.); jichenli666@163.com (J.L.); zhangyong8@ivdc.chinacdc.cn (Y.Z.); 2Henan Center for Disease Control and Prevention, Zhengzhou 450003, China; weiamy120@163.com; 3Hebei Center for Disease Control and Prevention, Shijiazhuang 050024, China; grace5257@126.com; 4Sichuan Center for Disease Control and Prevention, Chengdu 610044, China; tong-wen-bin@163.com; 5Shandong First Medical University & Shandong Academy of Medical Sciences, Jinan 271016, China; wangwh0508@126.com (W.W.); hycaven@126.com (Y.H.)

**Keywords:** coxsackievirus A14, phylodynamic analysis, recombination

## Abstract

As the proportion of non-enterovirus 71 and non-coxsackievirus A16 which proportion of composition in the hand, foot, and mouth pathogenic spectrum gradually increases worldwide, the attention paid to other enteroviruses has increased. As a member of the species enterovirus A, coxsackievirus A14 (CVA14) has been epidemic around the world until now since it has been isolated. However, studies on CVA14 are poor and the effective population size, evolutionary dynamics, and recombination patterns of CVA14 are not well understood. In this study, 15 CVA14 strains were isolated from HFMD patients in mainland China from 2009 to 2019, and the complete sequences of CVA14 in GenBank as research objects were analyzed. CVA14 was divided into seven genotypes A-G based on an average nucleotide difference of the full-length VP1 coding region of more than 15%. Compared with the CVA14 prototype strain, the 15 CVA14 strains showed 84.0–84.7% nucleotide identity in the complete genome and 96.9–97.6% amino acid identity in the encoding region. Phylodynamic analysis based on 15 CVA14 strains and 22 full-length VP1 sequences in GenBank showed a mean substitution rate of 5.35 × 10^−3^ substitutions/site/year (95% HPD: 4.03–6.89 × 10^−3^) and the most recent common ancestor (tMRCA) of CVA14 dates back to 1942 (95% HPD: 1930–1950). The Bayesian skyline showed that the effective population size had experienced a decrease–increase–decrease fluctuation since 2004. The phylogeographic analysis indicated two and three possible migration paths in the world and mainland China, respectively. Four recombination patterns with others of species enterovirus A were observed in 15 CVA14 strains, among which coxsackievirus A2 (CVA2), coxsackievirus A4 (CVA4), coxsackievirus A6 (CVA6), coxsackievirus A8 (CVA8), and coxsackievirus A12 (CVA12) may act as recombinant donors in multiple regions. This study has filled the gap in the molecular epidemiological characteristics of CVA14, enriched the global CVA14 sequence database, and laid the epidemiological foundation for the future study of CVA14 worldwide.

## 1. Introduction

Enterovirus is a nonenveloped, positive-stranded RNA virus belonging to the genus *Enterovirus* of *Picornaviridae* of the order *Picornavirales*, which is composed of 12 enterovirus (EV) A-L and 3 rhinovirus (RV) A-C species [1]. Human enteroviruses can be divided into *EV-A*, *EV-B*, *EV-C*, and *EV-D* [2], including poliovirus, coxsackievirus, echovirus, and the newly discovered enterovirus [3]. Coxsackievirus A14 (CVA14) belongs to the EV-A species, the genome of CVA14 is approximately 7.4 kb in length and consists of two noncoding regions and an open reading frame (ORF) in the middle. The two ends of the ORF are the 5′ untranslated region (5′UTR) and 3′ untranslated region (3′UTR), respectively. The ORF is composed of a structural protein coding region (P1) and two nonstructural protein coding regions (P2 and P3). P1 is cleaved into four capsid proteins including the VP1-VP4, P2, and P3 regions which are nonstructural protein coding regions that encode 2A, 2B, and 2C and 3A, 3B, 3C, and 3D, respectively [4].

Based on previous studies of EV-A species, we realize that CVA14 and enterovirus 71 (EV-A71) not only have high structural and sequence similarity in the capsid protein region but also rely on the same receptor to infect cells. When it comes to their capsid protein, CVA14, CVA16, and CVA7 show a high resemblance to EV-A71, sharing the same number of amino acid insertions and deletions in the VP2 and VP3 regions compared with others of species enterovirus A [5]. Likewise, as the VP1 region has neutralizing epitopes in the capsid protein region and plays a central role in the recognition of enterovirus [6], the amino acid similarity between CVA14 and EV-A71 in this region is second only to that between CVA16 and EV-A71. Furthermore, the SCARB2 receptor is an important vector for EV-71 infection, and CVA14 also relies on the SCARB2 receptor to invade cells. Both EV-71 and CVA14 infection depend on the SCARB2 pathway [7]. However, as the main pathogen causing severe hand, foot, and mouth disease (HFMD), EV-A71 is the mainstream direction of HFMD research at home and abroad. At present, EV-71 vaccines have been developed to address EV-71 epidemic cases around the world. However, global attention to CVA14 is low, and related research is very limited, so much research is needed for exploration.

Although CVA14 is not dominant in the pathogenic spectrum of enterovirus, it can still cause diseases such as HFMD, aseptic meningitis, and acute abdominal pain [8,9,10], and the epidemic regions are mainly distributed in Asia and Africa. In 1950, the prototype strain of CVA14 G-14 was isolated and obtained at the Medical Research Institute of South Africa [5]. Subsequently, CVA14-related clinical symptoms were reported in some countries and regions, such as Sweden and the United Kingdom [9,10]. In the previous study of Yao and Ma et al. [8,11], CVA14 was divided into three genotypes, A–C, which were used to construct a phylogenetic tree based on partial VP1 sequences of CVA14. According to a recent study that categorized CVA14 genotypes based on 15–25% nucleotide divergence in their complete VP1 sequences, the G genotype is the predominant genotype of CVA14 worldwide, and all Chinese strains currently belong to this genotype [12]. Further research of CVA14 phylodynamic and phylogeographic analysis was carried out to provide a reliable theoretical basis for the study of CVA14 genetic characteristics and virus evolution analysis.

Enteroviruses have three mechanisms including a high mutation rate, lack of correction function in the process of gene replication, and easy recombination, which lead to continuous evolution and spread. Recombination events also occur widely in human enterovirus [13], and recombination can reassemble genes with different dominant characteristics into a new genome, causing viral resistance, immune evasion, and virulence evolution [14]. In addition, statistical phylogenetic analysis of CVA14 by using the Bayesian stochastic search variable selection (BSSVS) model in BEAST software is of great significance in inferring and analyzing the spatial and temporal evolution of CVA14 [15]. Combining sequences with geographic information systems allowed the mapping of the temporal and spatial distribution of CVA14 across the globe. To date, no detailed molecular epidemiological study has revealed the transmission pattern and genotype distribution of CVA14 strains, and any phylogenetic data have been used to infer their geographical spread. In this study, we collected 15 CVA14 strains in mainland China and all the full-length sequences of CVA14 in GenBank and analyzed the molecular evolution and recombination of CVA14 strains.

## 2. Materials and Methods

### 2.1. Virus Isolation

We used human rhabdomyosarcoma (RD) cells and human laryngealcarcinom (Hep-2) cells to inoculate the sample, which was recorded as CVA14 in the Chinese HFMD surveillance network. RD cells and Hep-2 cells were supplied by the American Center for Disease Control and Prevention for virus isolation. We harvested viruses from the propagation of cell cultures until a complete cytopathic effect was observed. According to credible standards and strict protocols [16], viral RNAs were extracted from viruses using the QIAamp Viral RNA Mini Kit (QIAGEN, Hilden, Germany). All these viral RNAs were amplified by reverse transcription-polymerase chain reaction (RT-PCR) to obtain their complete VP1 capsid regions (888 nt) using the Prime Script One Step RT-PCR Kit Ver.2 (TaKaRa, Dalin, China) with previously designed primers (E486/E488) [17] and sequencing to verify their genotypes. Finally, a total of 15 samples were identified as the CVA14 genotype, which were collected from Beijing (*n* = 2), Tianjin (*n* = 1), Liaoning (*n* = 1), Henan (*n* = 5), Hebei (*n* = 2), Shaanxi (*n* = 1), Sichuan (*n* = 2), and Yunnan (*n* = 1) in mainland China from 2009 to 2019.

### 2.2. Whole-Genome Sequencing

Based on the primer walking method, we designed primers to amplify the remaining genome of viral RNAs of CVA14 using RT-PCR (Appendix A). The amplification program was as follows: 1 cycle of 30 min at 50 °C; 1 cycle of 2 min at 94 °C; 40 cycles of 30 s at 94 °C; 40 cycles of 30 s at 50 °C; 40 cycles of 1 min 20 s at 72 °C; 1 cycle of 7 min at 72 °C; and finally holding at 4 °C. The RT-PCR products were purified using a QIAquick PCR Purification Kit (QIAGEN, Hilden, Germany), and then amplicons were bidirectionally sequenced using an ABI 3130 Genetic Analyzer (Applied Biosystems, Foster City, CA, USA). Finally, we assembled each sequence from different amplicons using the Sequencher program (Version 5.4.5) (GenCodes, Ann Arbor, MI, USA) to obtain the whole genome of CVA14.

### 2.3. Dataset Construction

We obtained all of the CVA14 full-length VP1 sequences from the GenBank database by using “coxsackievirus A14” as an index word (dated 1 March 2023). To ensure the quality of sequences from the GenBank database was available, we eliminated low-quality sequences and evaluated the regression of root–tip distance to sampling time by using TempEst analysis [18]. A total of 22 complete VP1 sequences were finally obtained from the GenBank database including prototype strain G-14 (Appendix A). Finally, we selected 37 full-length VP1 sequences for dataset construction to conduct phylogenetic analysis and phylodynamic analysis. A total of 29 of 37 CVA14 full-length VP1 sequences were used for phylogeographic analysis of China. A total of 37 full-length VP1 sequences were used for worldwide phylogeographic analysis. We downloaded additional sequences of the EV-A group from the GenBank database for subsequent recombination analysis.

### 2.4. Phylodynamic Analysis

Based on 37 CVA14 full-length VP1 sequences (888 nt), we performed phylodynamic analysis using BEAST (version 1.10) [15] to estimate the mean substitution rate and the time of the most recent common ancestor (tMRCA) of CVA14 to explore and realize their evolutionary characteristics. First, TempEst analysis was used to verify the reliable temporal signal of CVA14 VP1 sequences by importing the sampling times of CVA14. Then, the asymmetric substitution model with the BSSVS option was selected to estimate the diffusion rates among sampling locations, and a strict clock model with a Bayesian skyline was used to estimate the effective population size of CVA14 at the same time. After that, we set the chain length to 80,000,000 to maintain sufficient running time for the procedure for a reliable running outcome. Finally, the above-mentioned settings and procedures were imported into BEAST. The outcome after running BEAST was analyzed within the TRACER program (version 1.7.1) [19] to check whether the effective sample size (ESS) was more than 200 [20] to ensure acceptance of convergence and reliable quality control parameters of the posterior distribution. With the first 10% of sampled trees removed using the burn-in option, the maximum clade credibility (MCC) tree was constructed using Tree Annotator (version 1.10.4), and the resulting tree was shown in FigTree (version 1.4.4). We obtained migration pathways of CVA14 by summing up the latitude and longitude of the sampling locations and the sampling times into a dataset and then importing it into the SpreaD3 package (version 0.9.7) together with the resulting log files to understand the spatial dynamics of CVA14. According to BF calculations resulting from the SpreaD3 package, the possible migration pathways of CVA14 were speculated in mainland China and the world. If the BF was greater than 3 and the posterior mean value was greater than 0.5, the migration pathways were considered to be significant and effective for visualizing the pathogen phylodynamic reconstruction among sampling locations [21].

### 2.5. Recombination Analysis

To detect the occurrence of recombination of the 15 CVA14 strains in this study, we constructed neighbor-joining (NJ) phylogenetic trees of the 5′UTR, P1, P2, and P3 regions using the sequences of EV-A prototype strains from GenBank and 15 CVA14 strains with 1000 bootstrap replicates as the first step.

After retrieving P2 and P3 regions of 15 CVA14 strains to BLAST to obtain the full sequences with more than 90% similarity in GenBank as the analysis objects, the Simplot program (version 3.5.1) was used to estimate recombination signal as the second step to lead compelling outcomes of recombination through similarity plots and bootscanning analysis. Using the sevens methods in Recombination Detection Program 4 (RDP4, version 4.46) was regarded as the final step, including RDP, GENECONV, Chimarea, MaxChi, BootScan, SiScan, and 3Seq with exploring the breakpoints and major and minor parents for checking and verifying the results of recombination analysis.

### 2.6. Base Substitution and Amino Acid Mutation Analysis

To further explore the differences between the 15 CVA14 strains and the CVA14 prototype strain, nucleotide and deduced amino acid sequences of 15 strains were compared with the CVA14 prototype strain using the BioEdit program (version 7.2.5.0). The antigenic sites of the coding region of the CVA14 prototype strain were predicted through an online website (IEDB.org: Free Epitope Database and Prediction Resource) [22]. To further study the antigenic site mutations of the 15 CVA14 strains, the amino acid sequences were compared between the 15 CVA14 strains and the CVA14 prototype strain.

### 2.7. Nucleotide Sequence Accession Numbers

Data were deposited in the China National Microbiology Data Center (NMDC) with accession numbers NMDCN0002PGQ-NMDCN0002PGV and NMDCN0002PH0-NMDCN0002PH8 (https://nmdc.cn/ accessed on 17 November 2023).

## 3. Results

### 3.1. Dataset Description

A total of 37 complete VP1 sequences worldwide were assembled for subsequent analysis, including 15 sequences from this study and 22 sequences downloaded from GenBank. For the sequences in this study which were isolated from eight provinces of China, their information was settled and listed (Table 1). Other sequences from GenBank whose sampling locations included China (*n* = 14), Russia (*n* = 1), India (*n* = 2), Madagascar (*n* = 3), Senegal (*n* = 1), and South Africa (*n* = 1). Of all 37 VP1 samples, 1 (OK570249 MAD-2011) [23] from Madagascar was isolated from the stool sample of a healthy child, 2 from India and 1 from Senegal were both from patients with AFP [24,25], and the others all from patients with HFMD or enterovirus surveillance, except for the prototype of CVA14 strain isolated from the institute of medicine of South Africa [5].

The complete genome nucleotide sequences of 15 CVA14 strains were 7410 to 7415 nucleotides in length with a consecutive ORF of 6579 nucleotides encoding a polypeptide of 2193 amino acids, with the addition of a noncoding 5′UTR with 750–755 nucleotides and a noncoding 3′UTR with 81 nucleotides. Each region of nucleotide and encoded amino acid of the 15 CVA14 strains was compared with the CVA14 prototype strain (Appendix A) to understand the detailed identity of each region. Additionally, the identity of the nucleotide and encoded amino acid of full-length genomic sequence compared with the CVA14 prototype strain were 84–84.7% and 96.9–97.6%, respectively. The identities of the nucleotides and encoded amino acids of full-length genomic sequence among the 15 CVA14 strains were 88.4–99.5% and 97.7–99.8%, respectively.

### 3.2. Phylodynamic Analysis

From the Bayesian skyline (Figure 1A), we realized that effective population size in the full-length VP1 sequences of CVA14 rose slowly from 1950 to 1980 and maintained a relatively maximal size steady without fluctuations until 2004. The effective population size began to rapidly rise after a phase with irregular decline between 2004 and 2013, confirming that CVA14 samples with a greater proportion than other years were collected in 2013. Later, a decline occurred with a short phase between 2013 and 2014, and no fluctuation has occurred until recently. Based on 37 full-length VP1 sequences, the maximum clade credibility (MCC) tree constructed with estimated divergence times of CVA14 was generally consistent with the phylogenetic tree in terms of topology and typing results, as shown in the MCC tree (Figure 1B). Meanwhile, the mean substitution rate and tMRCA were generally calculated by temporal phylogenies for the evolution of CVA14 full-length VP1 sequences. The results showed that the mean substitution rate was 5.35 × 10^−3^ substitution/site/year (95% HPD range 4.03–6.89 × 10^−3^) and the predicted tMRCA was approximately 1942 (95% HPD: 1930–1950). The earliest time of transmission in China estimated through BEAST analysis was 2005, which could also be verified through the node date of the CVA14 Chinese cluster shown in the MCC tree.

We reconstructed the spatial transmission patterns from 1950 to 2009 with six sampling countries (South Africa, Senegal, Madagascar, India, Russia, and China) based on 37 full-length VP1 sequences to explore and understand the global migration pathways of CVA14. The results of the phylogeographic analysis indicated that epidemic CVA14 was mainly concentrated in Asia and Africa and two possible migration pathways including South Africa to India and Russia to China (Figure 2) (support with BF ≥ 3; Appendix A). In addition, based on 29 full-length VP1 sequences isolated from provinces of mainland China including Shandong, Henan, Hebei, Beijing, Jiangsu, Yunnan, Sichuan, Shaanxi, Tianjin, and Liaoning, we analyzed the transmission path of CVA14 in mainland China through the same general method. The results showed that three possible migration pathways existed among sampling provinces: (1) Liaoning to Hebei; (2) Henan to Beijing; (3) Yunnan to Henan (supported by BF ≥ 3; Appendix A), strongly confirming the epidemic transmission of CVA14 in mainland China (Figure 3).

### 3.3. Recombination Analysis

Based on 15 sequences from this study and all sequences of EV-A prototype strains downloaded from GenBank, we constructed phylogenetic trees of the 5′UTR, P1, P2, and P3 regions to preanalyze the occurrence of recombination. In the 5′UTR phylogenetic tree (Figure 4A), we realized that the other 14 sequences were concentrated to become one cluster, except for the CHN_2015_SN_61 strain, which obviously formed one branch with CVA14 and coxsackievirus A4 (CVA4) prototypes, suggesting that the CHN_2015_SN_61 strain probably has a particular recombination signal in the 5′UTR. In the P1 region (Figure 4B), 15 sequences formed one cluster with the CVA14 prototype strain, which indicated that P1 is a conservative and stable region for encoding structural proteins without tending to undergo recombination. In the P2 region (Figure 4C), 15 sequences were concentrated in the cluster; however, coxsackievirus A5 (CVA5) and coxsackievirus A16 (CVA16) prototypes were closer to 15 sequences than the CVA14 prototype. In the P3 region (Figure 4D), a similar condition occurred in which the CVA16 prototype was closer to 15 sequences than CVA14. Therefore, we could conclude that the 15 strains from our study underwent recombination in the P2 and P3 regions with high probability, as different results appeared in phylogenetic trees.

According to all results of similarity plots and bootscanning analysis, it is noteworthy that four groups of recombination signals from the CHN_2009_HE_27, CHN_2013_HE_81, CHN_2013_HA_08, and CHN_2015_SN_61 strains and the remaining 11 strains lacked significant recombination signals. In the P1 region, the CHN_2009_HE_27 (Figure 5A), CHN_2013_HE_81 (Figure 5B), and CHN_2015_SN_61 (Figure 5D) strains all showed high similarity with the CVA14 group [8] which had been studied in previous papers in the P1 region as expected, except for the CHN_2013_HA_08 (Figure 5C) strain, which manifested very high similarity with the CVA14 prototype strain. For recombination signals in the P2 region, the CHN_2009_HE_27 strain had a relatively short range of recombination with the coxsackievirus A8 (CVA8) group in the 2C fragment of the P2 region. However, the CHN_2013_HE_81 strain had several visible fragments of recombination with the CVA8 group and the CHN_2013_HA_08 strain had very significant recombination with the CVA8 group in the entire P2 region. In particular, the CHN_2015_SN_61 strain with the coxsackievirus A2 (CVA2) group presented obviously high similarity in the 2C fragment of the P2 region. In the P3 region, the CHN_2009_HE_27 strain had several simple recombination signals with the CVA8 group until high similarity was observed in the terminus of the P3 region and 3′UTR. The CHN_2013_HE_81 strain showed relatively high similarity with the CVA4 and coxsackievirus A6 (CVA6) groups, indicating that CVA4 and CVA6 both more likely recombined with the CHN_2013_HE_81 strain in the P3 region. The CHN_2013_HA_08 strain had a similar recombination condition as the CHN_2013_HE_81 strain in the P3 region. The recombination signals of the CHN_2015_SN_61 strain with CVA2 and CVA4 groups were detected with high similarity in the P3 region. In particular, the CHN_2015_SN_61 strain with the CVA2 group had a very highly similar recombination fragment in their 5′UTR.

To analyze the detailed breakpoints of four groups of recombination signals using RDP4, at least three of the seven methods were judged as recombination and the *p*-value cutoff was 0.05, which was regarded as significant recombination [26]. The results (Table 2) showed that the breakpoint positions of the CHN_2009_HE_27 strain were mainly located at 3544-7372, covering the entire P2, P3, and 3′UTR regions. The major and minor parents were CVA14 (GenBank number: KP036482) and CVA8 (GenBank number: MT648783), respectively. The breakpoint positions of the CHN_2013_HE_81 strain were located at 4808-7138, including part of the 2C fragment and generally the entire P3 region. The major and minor parents were CVA14 (GenBank number: KP036483) and CVA4 (GenBank number: MK391065), respectively. We found 3343-4729 and 4777-7121 as the breakpoint positions of the CHN_2013_HA_08 strain, covering the P2 and P3 regions of the genome; CVA14 (GenBank number: AY421769) and CVA8 (GenBank number: KM609479) as the major and minor parents of the CHN_2013_HA_08 strain in the P2 region; CVA8 (GenBank number: KM609478) and CVA4 (GenBank number: MK391065) as the major and minor parents in the P3 region, respectively. The breakpoint positions of the CHN_2015_SN_61 strain were located at 39-612 and 3884-7243, covering 5′UTR, part of the P2 region, and the entire P3 region. The major and minor parents were CVA14 (GenBank number: KP036483) and CVA2 (GenBank number: KP289358) in 5′UTR, respectively, and the major and minor parents were CVA14 (GenBank number: AY421769) and CVA2 (GenBank number: JX867332) in the P2 and P3 regions, respectively. The above-mentioned results of RDP4 are generally similar to similarity plots and bootscanning analysis, confirming the higher reliability of recombination analysis.

### 3.4. Base Substitution and Amino Acid Mutation Analysis

Lack of function of proofreading in virus replication leads to enterovirus tending to mutate in the entire genome [27]. A total of 15 sequences contained 1132-1183 nucleotide mutations in full-length genome and 51-66 amino acid mutations in their coding regions compared with the CVA14 prototype strain. A total of 27 common mutation sites were found in amino acid mutations, including 1 acidic amino acid replaced by basic amino acid, 1 nonpolar amino acid replaced by polar amino acid, and 3 polar amino acids replaced by nonpolar amino acids. A total of 15 CVA14 sequences had nucleotide deletions at sites 109, 110, and 111. Likewise, there were 30-858 nucleotide differences in the full-length genome and 3-50 amino acid differences in coding regions among the 15 sequences. The possible antigenic sites of CVA14 prototype strain were predicted by using tools on an online website for exploring the relationship of amino acid mutations with antigenic sites of CVA14 and laying the foundation for further study of CVA14. The CVA14 prototype strain probably contained 7-8 antigenic sites in the VP1-VP3 regions (Table 3). A total of 11 common antigen mutation sites were found in 15 sequences by comparing their differences with the CVA14 prototype strain: [VP1: I→T(23), E→G(39), Y→S(61), L→M(62), A→T(241), K→T(291); VP2: Q→E(72), F→S(161), T→S(227); VP3: E→K(59), N→S(63)] [28]. Some number of mutations in enterovirus evolution are significant for their propagation, pathogenesis, and dissemination [29,30,31]. Unresolved problems of mutations in CVA14 deserve to be studied to understand their role in the evolution of CVA14 and enrich the relative information in enteroviruses.

## 4. Discussion

In recent years, the prevalence of enteroviruses dominated by non-EV-A71 and non-CVA16 has gradually increased at home and abroad, and their reports in HFMD cases have increased significantly, such as CVA6, CVA10, and CVA4 [32,33,34,35,36]. Especially in many regions of mainland China, other enteroviruses once surpassed EV-A71 and CVA16 to become the dominant serotype becoming the main pathogen in the pathogenic spectrum of HFMD [37,38]. Therefore, the surveillance and research of other enteroviruses are becoming more and more important. CVA14 as a member of species A (EV-A) has distinct differences in clinical symptoms and disease severity from other EV-A viruses. The reported cases and symptoms caused by CVA14 mainly include HFMD, aseptic meningitis, and acute abdominal pain [8,9,10], which, when caused by it, are self-limited [9,10].

According to the image of the Bayesian skyline, we can roughly understand that the change characteristics of the effective population size of CVA14 were relatively stable without much fluctuation before 2004, indicating that the possibility of CVA14 causing large-scale epidemic and spread in the world was relatively small during this period. However, the effective population size of CVA14 has had some significant fluctuation since 2005, and it did not reach a stable state until 2015. These changes were consistent with the epidemiological data we collected. The MCC tree constructed based on 37 full-length VP1 sequences of CVA14 was basically consistent with the branch composition of the reported phylogenetic tree [12]. Previous studies classified CVA14 into seven genotypes, A–G, based on the average nucleotide difference of the VP1 coding region of more than 15%. The MCC tree constructed in this study shows the time divergence nodes of each branch, to better understand and analyze the characteristics and trends of CVA14 evolution over time. Except for the two CVA14 strains isolated in India, which were divided into two different genotypes, the CVA14 strains isolated in the same area were basically divided into the same genotype. Genotype G was the main genotype circulating worldwide, which was concentrated in the mainland of China. It is known that the first CVA14 strain in the mainland of China was isolated in Shandong province in 2006 (GenBank number: GQ253379). Since then, the G genotype began to circulate and spread in the mainland of China forming a multiprovincial and interprovincial transmission chain. By analyzing the phylogeographic results, we obtained a continuous and reliable migration path of CVA14 around the world during the past six decades. The results showed that the CVA14 prototype strain (G-14) was isolated in 1950 at the Institute of Medical Research, Republic of South Africa [5], and then spread to India, Russia, and China (possible support with BF ≥ 3). In addition, in mainland China, Yunnan, Henan, Hebei, Liaoning, and Beijing, it existed in neighboring provinces and interprovincial migration paths (BF ≥ 3) indicating that although the prevalence of CVA14 was lower than other dominant serotypes, it could still form a relatively obvious small-scale transmission at home and abroad, so relevant monitoring and research are still needed. The above pattern of phylogeographic analysis may be influenced by missing epidemiological data, and due to the imperfect surveillance system for enterovirus in some regions and the lack of related research on CVA14, the full-length VP1 sequences of CVA14 available for analysis in GenBank are very limited. The epidemiological characteristics of CVA14 in the mainland of China were representative, but the epidemiological characteristics of CVA14 in the world need to be more comprehensive and further studied. In addition, due to the differences in sampling time and location, the collected data may have bias on the results of analysis.

Enterovirus has two mechanisms including frequent mutation and frequent recombination, which enable it to adapt to the new environment quickly and maintain the stability of its survival and evolution. Therefore, the study of the mechanism of mutation and recombination of enterovirus is an important measure to analyze enteroviruses [28,39]. Based on the phylogenetic tree of the P2 region, it is not hard to see that the CHN_2009_HE_27, CHN_2013_HE_81, CHN_2013_HA_08, and CHN_2015_SN_61 strains are close to the CVA5 prototype strain. In the phylogenetic tree of the P3 region, these four strains are very close to the CVA16 prototype strain. These results suggested that the CHN_2009_HE_27, CHN_2013_HE_81, CHN_2013_HA_08, and CHN_2015_SN_61 strains might have different recombinations during evolution. Four groups of recombination patterns were identified by Simplot and RDP4. Among the four groups of recombination patterns, CVA2, CVA4, CVA6, CVA8, CVA12, and CVA14 were obtained from other regions [8] and the CVA14 prototype strains were speculated as possible parents. CVA4 and CVA6 are the predominant serotypes circulating in China and abroad at present. Whether the recombination of CVA14, CVA4, and CVA6 will cause a change in the transmission ability needs further exploration in virology, to do basic research on the recombination events occurring in enterovirus with low prevalence and high prevalence.

We predicted the epitopes of the structural proteins of the CVA14 prototype strain and found that the major epitopes were concentrated in the VP1-VP3 region. For the VP1 region, which plays a role in the process of enterovirus recognition, adsorption, entry into target cells, and virion assembly, compared with the prototype CVA14 strain, the 15 CVA14 strains in this study had 6 amino acid common mutations in the possible epitopes of this region. Whether these mutations are related to the static epidemic of CVA14 in mainland China and whether they are beneficial to the evolution and transmission of CVA14 needs further study. Therefore, the surveillance of enterovirus worldwide should be strengthened and studied more carefully and deeply. A total of 15 HFMD-associated CVA14 strains isolated from the National Polio Laboratory were used in this study, which enriched the global CVA14 sequence database. We analyzed the recombination pattern of CVA14 with other enteroviruses, explored the molecular epidemic characteristics of CVA14 in mainland China and around the world, and predicted the amino acid mutation sites associated with CVA14, which deepened our understanding of CVA14 and provided valuable information for the molecular epidemiology of CVA14. With the increasing complexity of the pathogen spectrum of HFMD, the proportion of the original dominant strains CVA16 and EV-A71 decreased, and the proportion of other enterovirus strains increased. At the same time, there were some rare, easily ignored, and vulnerable epidemic serotypes of enterovirus cocirculating. Therefore, the surveillance and reporting of other non-EV-A71 and non-CVA16 enteroviruses should be strengthened to fully understand the global epidemic trend of HFMD.

## Figures and Tables

**Figure 1 viruses-15-02323-f001:**
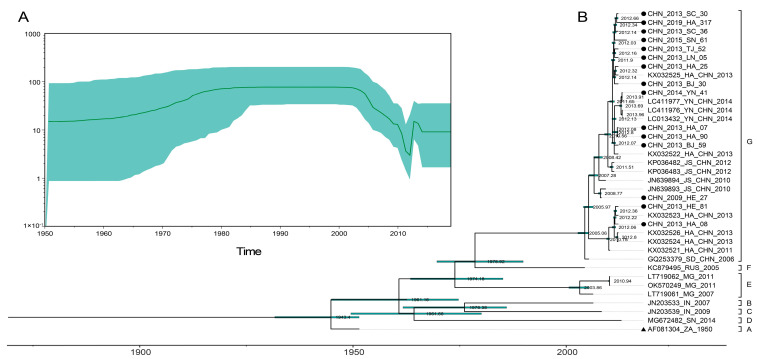
(**A**) Bayesian skyline plot of the worldwide full-length VP1 sequences of CVA14 strains. (**B**) The phylogenetic tree was constructed with maximum clade credibility (MCC) for 37 VP1 sequences of CVA14 strains. The green bar indicates 95% HPDs of tMRCAs. ● represents 15 CVA14 strains from this study, ▲ represents the CVA14 prototype strain (G-14).

**Figure 2 viruses-15-02323-f002:**
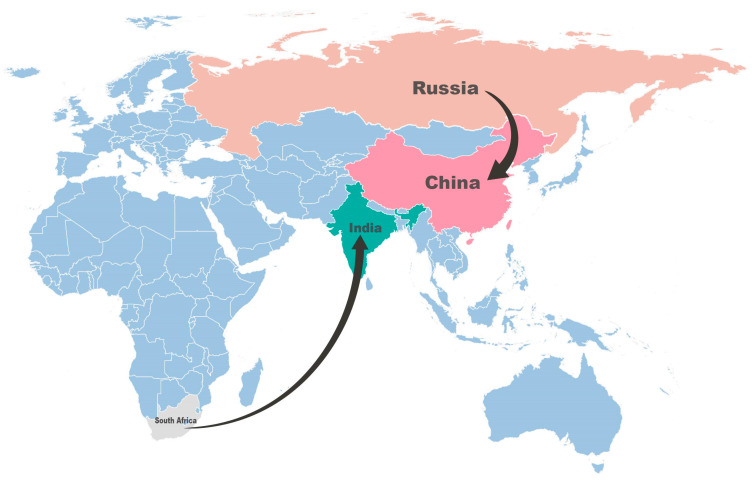
Spatial diffusion pathways that are shown with supported BF ≥ 3 and posterior mean value ≥ 0.5 among worldwide sampling countries.

**Figure 3 viruses-15-02323-f003:**
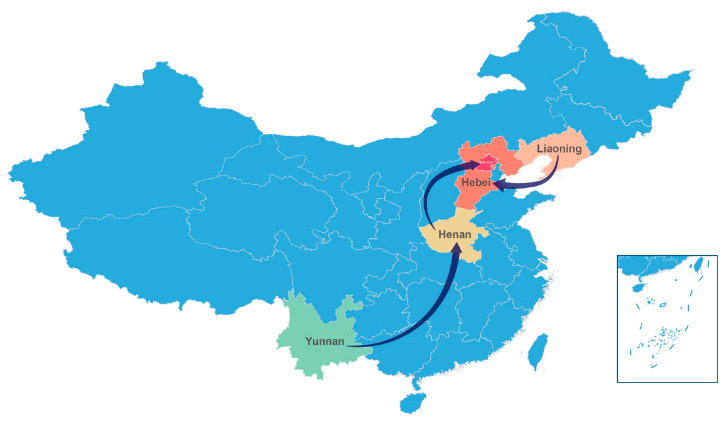
Spatial diffusion pathways are shown with supported BF ≥ 3 and posterior mean value ≥ 0.5 among sampling countries in mainland China.

**Figure 4 viruses-15-02323-f004:**
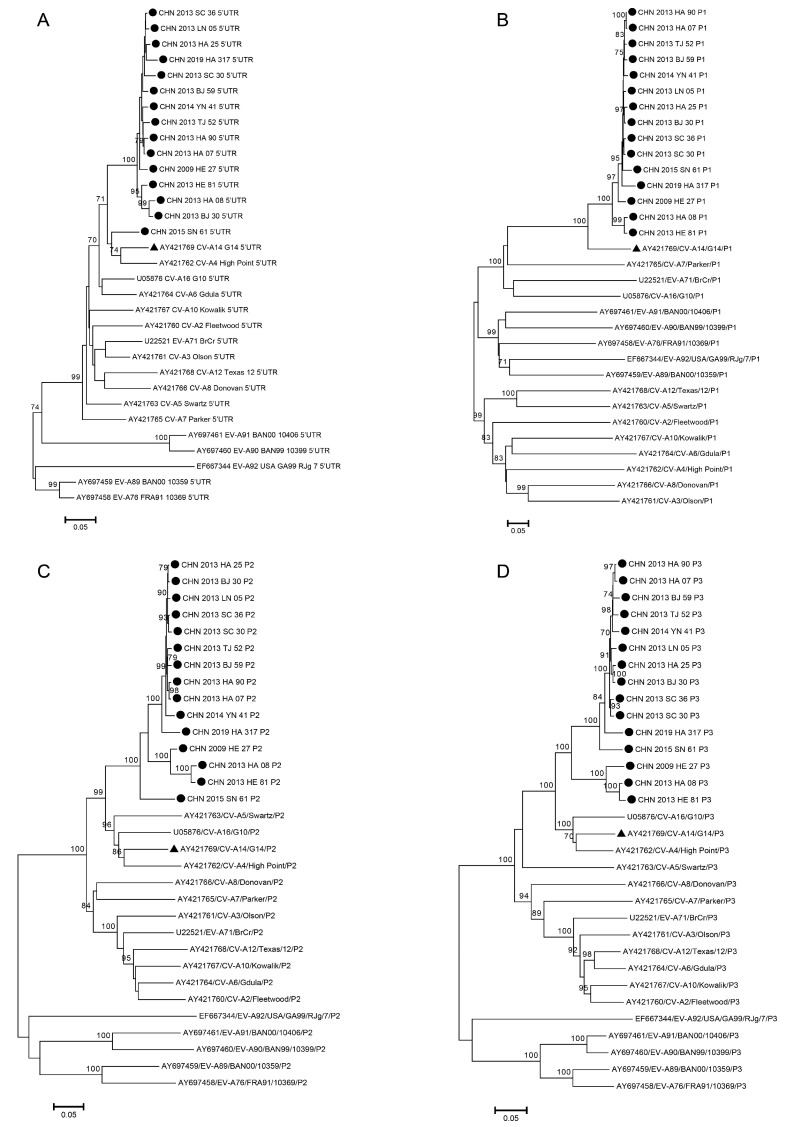
Neighbor-joining phylogenetic trees based on 5′UTR (**A**), P1 (**B**), P2 (**C**), and P3 (**D**) of the 15 CVA14 strains from this study and EV-A prototype strains. The numbers on the codes represent the bootstrap support of the node (1000 bootstrap replicate percentage). The scale bars represent the replacement rate at each site per year. ● represents 15 CVA14 strains from this study, ▲ represents the CVA14 prototype strain (G-14).

**Figure 5 viruses-15-02323-f005:**
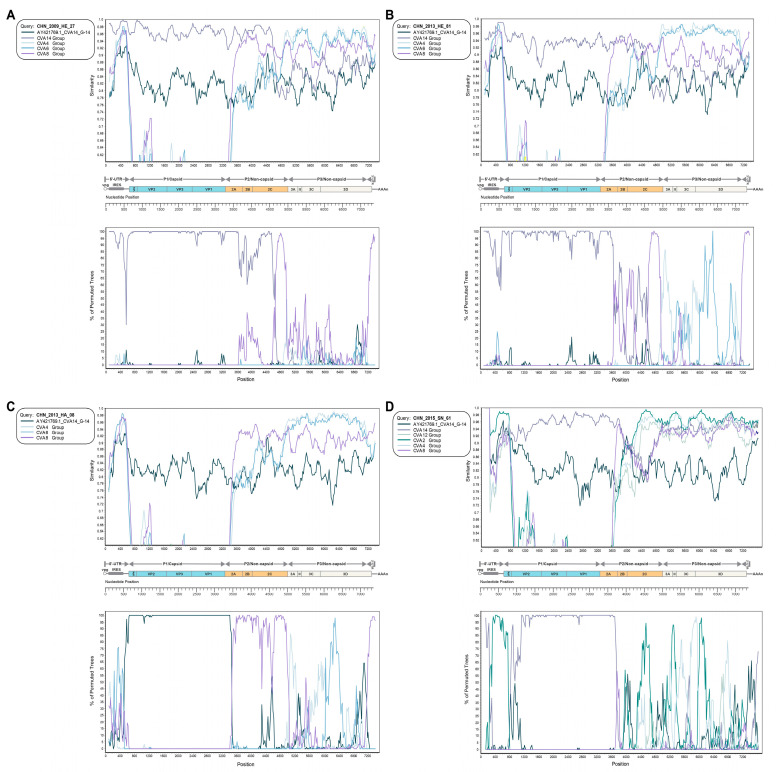
Recombination events from (**A**) CHN_2009_HE_27, (**B**) CHN_2013_HE_81, (**C**) CHN_2013_HA_08, and (**D**) CHN_2015_SN_61 strains were shown through similarity plots and bootscanning analyses.

**Table 1 viruses-15-02323-t001:** Relevant information for the 15 CVA14 strains in this study.

Name	Province	Gender	Age	Date of Sample	Specimen Type	Case Classification
CHN_2009_HE_27	Hebei	Male	3	20 April 2009	Rectal swab	Severe
CHN_2013_BJ_30	Beijing	Female	13	15 May 2013	Nasopharyngeal swab	Mild
CHN_2013_BJ_59	Beijing	Male	3	13 June 2013	Nasopharyngeal swab	Mild
CHN_2013_HE_81	Hebei	Male	4	8 April 2013	Rectal swab	Mild
CHN_2013_HA_07	Henan	Male	4	14 April 2013	Stool	Mild
CHN_2013_HA_08	Henan	Female	3	3 May 2013	Stool	Mild
CHN_2013_HA_25	Henan	Female	5	5 May 2013	Stool	Mild
CHN_2013_HA_90	Henan	Male	2	31 March 2013	Stool	Mild
CHN_2013_LN_05	Liaoning	Male	1	20 March 2013	Stool	Mild
CHN_2013_SC_30	Sichuan	Female	1	3 June 2013	Nasopharyngeal swab	Mild
CHN_2013_SC_36	Sichuan	Male	7	8 May 2013	Nasopharyngeal swab	Mild
CHN_2013_TJ_52	Tianjin	Female	<1	5 June 2013	Stool	Mild
CHN_2014_YN_41	Yunnan	Female	2	18 February 2014	Stool	Mild
CHN_2015_SN_61	Shaanxi	Male	8	1 June 2015	Nasopharyngeal swab	Mild
CHN_2019_HA_317	Henan	Female	2	5 April 2019	Stool	Mild

**Table 2 viruses-15-02323-t002:** Characteristics of four recombination patterns of Chinese CVA14.

Strain	Breakpoint Positions	Covering Regions	Recombinant Donors
CHN_2009_HE_27	3544-7372	P2, P3, 3′UTR	Major: CVA14 (KP036482) Minor: CVA8 (MT648783)
CHN_2013_HE_81	4808-7138	P2, P3	Major: CVA14(KP036483) Minor: CVA4 (MK391065)
CHN_2013_HA_08	3343-4729	P2	Major: CVA14(AY421769) Minor: CVA8 (KM609479)
	4777-7121	P3	Major: CVA8 (KM609478) Minor: CVA4 (MK391065)
CHN_2015_SN_61	39-612	5′UTR	Major: CVA14(KP036483) Minor: CVA2 (KP289358)
	3884-7243	P2, P3	Major: CVA14 (AY421769) Minor: CVA2 (JX867332)

**Table 3 viruses-15-02323-t003:** The antigen sites of the CVA14 prototype strain (AY421769.1/G-14) predicted using online websites.

Region	Position	Predicted Antigen Epitope
VP1	19-75	LTSPIQTPTAANTNVSNHRIELGEVPALQAAETGATSLVSDEYLIETRCVVNSHSTE
	96-106	LQGTVNTGGFA
	143-144	GE
	160-173	PKPTGRNTYEWQTA
	208-219	PTFGKHLPADDF
	241-242	AP
	266-292	RSQPYVAKNYPNYKGSEIKCASSSRKS
VP2	6-25	ACGYSDRVAQLTIGNSTITT
	39-60	PEYCSDTDATAVDKPTRPDVSV
	69-79	KDWQASSKGWY
	84-87	DVLA
	132-166	TGTIAGNTGNEHTHPPYATTQPGLDGFPLFNPYVL
	203-208	FDSALN
	225-228	ASTA
	246-251	AGLRQA
VP3	5-38	ELPGTNQFLTTEDGTSAPILPGFHPTQVIHIPGE
	55-64	NNLESNENDP
	76-80	SEKGK
	88-95	DPGLDGPW
	138-147	GGVTPASRMD
	176-186	YRAQSKNQYFD
	205-207	AET
	230-238	DADSLTQTA

## Data Availability

The datasets used and analyzed in this study are obtained and available from the corresponding authors upon a reasonable request.

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
