# Peer review of "Molecular Epidemiology and Evolution of Coxsackievirus A14"

_viruses, 2023, doi:10.3390/v15122323_

Round 1
Reviewer 1 Report
Comments and Suggestions for Authors
In this manuscript, the authors examine the evolution of Coxsackievirus A14 by sequencing 15 isolates collected from China. The authors analyzed the VP1 gene in CVA14 and compared the sequences with other VP1 sequences in GenBank.
Major Concerns:
1. Genbank accession numbers for the analyzed sequences or the newly sequenced isolates were not provided. Further, the dates at which each individual sequence was collected were not provided, which is surprising since the authors try to suggest transmission and evolutionary patterns of CVA14.
2. Figure 2 and Figure three are “Spatial diffusion pathways,” which the authors say are supported by BF>3. Based on the text in the results, it is unclear how these data were even performed. Are there statistics to back up the likelihood of the proposed spatial diffusion of CVA14 into China?
3. The authors present their findings with minimal context of how the mutation rate and recombination impact CVA14 evolution. Therefore, it is hard for the reader to understand the importance of the results. For example, the authors present data for the antigenic sites for the prototype CVA14 strain. However, there is no analysis of how these antigenic sites are changing compared to the other sequences provided in the paper that might be useful for analyzing how the virus is evolving to evade the immune response.
Minor Concerns:
4. The English and grammar throughout the paper need to be addressed.
5. Numerous references need to be added to the text.
Comments on the Quality of English LanguageThe English and grammar throughout the paper need to be addressed.
Reviewer 2 Report
Comments and Suggestions for Authors
CVA14 is a rarely detected and neglected pathogen of hand, foot, and mouth disease and its genomic data in GenBank database is in scarce. The authors sequenced the complete genomes of 15 CVA14 strains, and analyzed the molecular epidemiology and evolutionary genetics of CVA14. Generally, the manuscript is well organized and the work is valuable. Minor revisions are need.
1. Some English and typos should be revised.
- Abstract, “84%-84.7% should be changed to “84.0%-84.7%”.
- In “10-3”, the “-3” should be superscript.
- “enterovirus group A” should be “species Enterovirus A”
- Introduction, the taxonomy name should be in italics. Such as “Picornavirales”, “EV-A”, etc. Notice: “Picornavirales”, not “Picornavirals”.
- Discussion. “Enterovirus has two mechanisms including easy mutation and easy recombination, ……”. This sentence should be revised.
2. Table 1. Regarding “/, the patient was less than one year old”, I suggest remove this note and change “/” to “< 1” in the table.
3. Legend to Figure 1. “VP1 CVA14 strains” be changed as “VP1 sequences of CVA14 strains”.
Comments on the Quality of English LanguageMinor revision of English language is needed.
Reviewer 3 Report
Comments and Suggestions for Authors
The authors report the molecular epidemiological study of CAV14 strains isolated under hand-foot-and-mouth disease surveillance. The authors also used a Bayesian method to estimate the divergence year using nucleotide sequence data of Chinese isolates and examined the point of the importation of CAV14 strain to China.
As the authors point out, information on CAV14 is limited, so this paper is informative for understanding the nature of CAV14.
Minor comments.
Material and method
Were all CAV14 strains in China isolated by RD-A? Please clarify it.
Result
CAV14 isolates in 2013 accounted for 11/15 of the total numbers have been used for analysis, but why so many 2013 strains?
And would there be any bias in the further analysis? Please clarify these points.
Discussion
The first strain was isolated in 2006 and the estimated divergence year is 2005.
Is this true? Could it be simply because CAV14 had not been found through examination under any surveillance activity. I would like to ask the author's y views?
After the event of recombination, these recombinants have become fixed in the population or not? I would like to ask the author's views?
Reviewer 4 Report
Comments and Suggestions for Authors
This study analyzes the sequences of coxsackievirus A14 (CVA14), a type of enterovirus A. The authors have used 15 isolates taken from 2009-2019 from several provinces of China and 22 sequences from GenBank over a period extending back to 1950 and from across the world. The authors have done phylogenetic and recombination analyses to define the relationship of these isolates to other CVA14 and other enterovirus A types to demonstrate sufficient nucleotide and amino acid identity to define their serotype and clade classification. All the new CVA14 isolates belong to the G clade along with a number of the CVA14 sequences in GenBank. There is some evidence of the spatial transmission pathways of CVA14 between Chinese provinces but the limitation of the number of isolates means that the suggestion of transmission between countries is unlikely to be informative as it is likely that these viruses in countries other than China are not represented sufficiently in the GenBank database. This should be noted in the text in which Figure 2 is discussed. Have these new Chinese CVA14 sequences been submitted to GenBank? If so, the accession numbers should be given. As the authors are using the sequences submitted to GenBank by others, it would be appropriate to reciprocate by providing their sequence data as well.
The recombination analysis is certainly supported by the comparison of the RDP4 results with the similarity plots and boot scanning analysis but would be much clearer if the Chinese genomes were shown in a table with the likely recombination sites and closest parental genomes.
In addition, there are a number of reference changes or missing references that should be addressed:
In “Likewise, as the VP1 region has neutralizing epitopes in the capsid protein region and plays a central role in the recognition of enterovirus (6)...”, Reference 6 is not appropriate. Huang KA. Structural basis for neutralization of enterovirus. Curr Opin Virol. 2021 Dec;51:199-206, a recent review on this topic, would be more to the point.
In “Recombination events also occur widely in human enterovirus (13),...” reference 13 is not a review of recombination in enteroviruses. There are good references for this:
Kriakopoulou Z., Pliaka, V., Amoutzias, G.D. et al. Recombination among human non-polio enteroviruses: implications for epidemiology and evolution. Virus Genes 50, 177-188 (2015) or Simmonds P., Welch J. Frequency, and dynamics of recombination within different species of human enteroviruses. J Virol. 2006 Jan; 80(1):483-93 would be more appropriate.
In “The possible antigenic sites of CVA14 prototype strain were predicted by using tools on an online website for exploring the relationship of amino acid mutations with antigenic sites of CVA14 and laying the foundation for further study of CVA14.” and in the caption for Table 2: “The antigen sites of CVA14 prototype strain (AY421769.1/G-14) predicted through online websites.”; the authors must give references for these online tools and website.
Comments on the Quality of English Language
In “Enterovirus has two mechanisms including easy mutation and easy recombination, which enable it to adapt to the new environment quickly and maintain the stability of its survival and evolution.” it would be better to use the word “frequent” than “easy.”
There are a few typographic errors as well.
